# Recovery of End-of-Life Tyres and Mineral Wool Waste: A Case Study with Gypsum Composite Materials Applying Circular Economy Criteria

**DOI:** 10.3390/ma16010243

**Published:** 2022-12-27

**Authors:** Alicia Zaragoza-Benzal, Daniel Ferrández, Paulo Santos, Carlos Morón

**Affiliations:** 1Departamento de Tecnología de la Edificación, Escuela Técnica Superior de Edificación, Universidad Politécnica de Madrid, 28040 Madrid, Spain; 2Institute for Sustainability and Innovation in Stuctural Engineering (ISISE), Department of Civil Engineering, University of Coimbra, 3030-788 Coimbra, Portugal

**Keywords:** gypsum composites, end-of-life tyre, recycled fibres, circular economy, precast

## Abstract

The building sector is currently undergoing a process of change due to concerns about the sustainability of the construction industry. The application of circular economy criteria to develop new, more sustainable construction products has become one of the major challenges for the society of the future. This research advances towards the development of new lightened gypsum composites that incorporate waste from end-of-life tyres and recycled fibres from mineral wool thermal insulation in their composition. The results show how it is possible to reduce the consumption of the original raw materials by replacing them with recycled rubber granular particles, developing new construction products that are lighter, with better water resistance and greater thermal resistance. Additionally, it is shown that the incorporation of recycled fibres from rock wool and glass wool insulation is a good solution to improve the mechanical resistance of lightened gypsum composites, giving these construction and demolition wastes a second useful life by reincorporating them in the process of manufacturing new prefabricated housing products.

## 1. Introduction

The construction sector generates a strong environmental impact because of the high energy consumption and CO_2_ emissions linked to its activity, as well as demanding large quantities of raw materials and producing high volumes of waste during the management of the building process [1,2]. The European Union has become the region leading international efforts to conserve natural resources and reduce global warming [3], setting as a roadmap the European Green Deal, which aims to serve as a reference framework for the transformation of the Union’s economy towards a more sustainable future [4]. In this context, the efficient use of resources in construction and the development of new construction materials under circular economy criteria have become the key points that allow sustainable development and economic growth to be combined [5].

Currently, numerous research projects are being carried out with the aim of developing new gypsum compounds for its application in construction, since these materials require less energy in their manufacturing process compared to other binders [6]. Nevertheless, gypsum has been used in construction since ancient times because of its excellent technical performance [7]. It is a non-combustible material, with a high capacity for hygrothermal regulation, excellent adhesion to ceramic surfaces, easy workability, and great versatility for making prefabricated systems [8,9]. However, these properties can be improved and have been the subject of extensive research in recent decades.

One of these lines of research has been linked to the incorporation of waste from end-of-life tyres (ELT). The consumption of tyres in the EU is approximately 3.6 million tonnes [10], which, in turn, generates large volumes of solid waste, causing a strong environmental impact. This waste accumulates in landfills in large volumes, emits high amounts of CO_2_ when incinerated, has a very slow degradation process, and is very harmful to the marine environment when dumped on coasts [11,12]. However, the recycling process generates by-products that are of great interest for their application as additives in gypsum composites. In this sense, the rubber granules generated after tyre shredding can be added as a partial replacement of the binder in prefabricated gypsum boards and panels, thus, reducing their thermal conductivity and improving their acoustic behaviour [13]. Nevertheless, these rubber aggregates lead to a decrease in the mechanical properties of gypsum and plaster composites [14], as well as to an increase in their porosity and water absorption capacity [15]. These effects can be partly corrected by the incorporation of synthetic fibres as reinforcement material [16], although at the cost of increasing the price of the final product. On the other hand, properly treated textile fibres from ELTs have allowed the development of gypsum composites with good acoustic absorption, without significant reductions in their mechanical properties and with good adhesion at the dihydrate–fibre interface [17].

Another of the major lines of research promoted in recent decades is the incorporation of construction and demolition waste (CDW) under circular economy criteria for the development of more sustainable gypsum composites [18]. CDW accounts for more than 35% of the total solid waste generated in the EU, and it is necessary to study new alternatives that allow its recovery, revalorisation, and reincorporation into the manufacturing process of new materials [19]. These CDWs include thermal insulation waste from mineral wool, classified as non-hazardous waste, but with a slow degradation process [20]. Some studies show how it is possible to incorporate rock wool or glass wool insulation waste into the manufacturing process of masonry mortars, improving their thermal behaviour and flexural strength [21]. Similarly, Romaniega-Piñero et al. developed their research by adding this type of recycled fibres to gypsum composites with the aim of replacing the commercial fibres used in the manufacture of prefabricated slabs (polypropylene and glass), obtaining good results in comparison with traditional gypsum composites that increased the flexural strength [22]. This type of recycled fibre was previously mixed with the gypsum powder and then the mixing process was started with water, achieving a good dispersion of the fibres in the binder matrix. It was observed that with fibre additions close to 1% by weight, it was possible to improve the thermal and mechanical performance of suspended ceiling panels [23].

The aim of this work is to study the technical feasibility of a novel gypsum composite material with the incorporation of granular particles from ELTs as a partial replacement of the original raw materials and with the addition of recycled fibres from mineral wool thermal insulation waste. The intention is to obtain a sustainable building material with high thermal performance, light weight, and good mechanical strength. To this end, an experimental campaign is proposed to carry out a physical–mechanical characterisation, as well as to study the possibilities of using this material to produce a prototype prefabricated building block.

## 2. Materials and Methods

This section describes the materials used to make the different mixes used in this research, as well as the dosages used and the experimental programme that was carried out.

### 2.1. Materials

Figure 1 shows the raw materials used to produce the lightened gypsum composites developed.

#### 2.1.1. Binder

The binder used for the mixing of the composites was fine construction gypsum, type YF-B1 according to UNE-EN 13279-1, supplied by the company Saint-Gobain Placo (San Martín de la Vega, Spain). It is a material commonly used for cladding and surface finishing of interior walls in dwellings, as well as to produce plates for false ceilings and prefabricated panels for interior partition walls [24]. Equation (1) shows the basic reaction scheme for obtaining this raw material from gypsum stone:(1)$${\mathrm{CaSO}}_{4}\cdot 2H_{2}O\;\to {\;\mathrm{CaSO}}_{4}\cdot \frac{1}{2}H_{2}O\;\left(\alpha ,\;\beta \right)+\frac{3}{2}H_{2}O$$

Firstly, Figure 2 shows the X-ray diffraction analysis of the powdered fine gypsum raw material, where the main diffraction peaks showing its crystalline structure are collected. The equipment used was Siemens Krystalloflex D5000 (Madrid, Spain) with a Cu-Kα graphite monochromator. The analysis was carried out in the interval range 0° < 2θ < 60°, obtained every 0.04° and a speed of one second per step.

In addition, Table 1 shows the main physical characteristics of fine gypsum YF-B1 obtained in accordance with standard UNE-EN 13279-1:2009 [25] and provided by the company supplying this binder.

#### 2.1.2. Recycled Crumbed Rubber

Rubber aggregates from end-of-life tyres (ELT) were used as an additive to lighten the gypsum composite prefabricated products produced in this research. This granular raw material was supplied by the organisation SIGNUS ECOVALOR S.L. (Madrid, Spain). The incorporation of these recycled raw materials in the process of manufacturing industrial prefabricated products enables the revaluation and recovery of waste tyres [26]. This means adding a high added value that allows the differentiation of the developed products with a marked environmental character [27]. Table 2 shows the physical properties obtained for the rubber aggregates used in this research. 

#### 2.1.3. Fibres from Thermal Insulation Waste

As a reinforcement material to improve the mechanical strength of the gypsum composites produced, manually cut 12 mm long thermal insulation mineral wool fibre residues were used. The main composition of these fibres is SiO_2_ (>60%), CaO (>20%), and other minority oxides such as Fe_2_O_3_, Na_2_O, K_2_O, Al_2_O_3_, SO_3,_ and TiO_2_ [28]. Figure 3 shows the cylindrical morphology of the shredded textile fibre waste obtained by scanning electron microscopy.

On the other hand, Table 3 shows the results derived from the physical characterisation of the thermal insulation mineral wool waste used.

#### 2.1.4. Water

The water used for the mixing of the gypsum compounds elaborated was drinking water from the Canal de Isabel II of the Community of Madrid (Spain). This water is of a harmless nature and has been successfully used in previous research work [28]. Its main characteristics include its medium hardness (25 mg CaCO_3_/l) and its pH between 7 and 8.

### 2.2. Sample Preparation

The preparation of the samples used in this research was carried out following the procedure of the UNE-EN 13279-2:2004 standard [29]. This standard describes the mixing process shown in Figure 4.

For the nomenclature of the different mixes produced in this research, the following structure was used: G0.7–rubber–insulation, where G0.7 refers to the water/gypsum ratio by mass, rubber refers to the amount of granulated rubber granules from end-of-life tyres added, and, finally, insulation refers to the type of thermal insulation residue recovered, which can be either glass wool (GW) or rock wool (RW).

The mass proportions used for the preparation of the different mixes are shown in Table 4, all of which were made following the same techniques and methods.

In all the mixes listed in Table 4, both the waste from end-of-life tyres and the fibres from mineral wool thermal insulation waste were previously mixed dry with the gypsum powder before starting the mixing. The samples were compacted by lifting the mould from one end by 10 mm and dropping it, repeating this process up to five times. Once the samples were set, they were removed from the moulds and stored at a temperature of 22 ± 2 °C and an ambient relative humidity of 60 ± 5%. After seven days of storage under laboratory conditions, the samples were dried in an oven at a constant temperature of 40 ± 2 °C for 24 h prior to testing.

### 2.3. Experimental Programme

In this research, an experimental campaign divided into three phases was developed. Firstly, a physical characterisation of the gypsum composite materials produced was conducted. Next, a mechanical characterisation was carried out to determine the resistance of these new materials, and, finally, in the last phase, the suitability of these gypsum composites to produce pieces that can be used in the construction of prefabricated partitions was analysed. 

In terms of physical characterisation, tests were carried out to determine the following properties: bulk density, thermal conductivity (Figure 5a), surface hardness (Figure 5b), capillary water absorption, and water vapour permeability (Figure 5c).

To obtain the surface hardness and bulk density values, the recommendations of the UNE 102042: 2014 standard were followed [30]. These physical properties were determined in series of three standardised RILEM specimens of 40 mm × 40 mm × 160 mm, using a Shore C hardness tester and a digital laboratory scale with an accuracy of 0.01 g;The thermal conductivity coefficient of the gypsum composites produced was obtained by means of the “Thermal Box” test. For this purpose, a thermal box was used with replaceable walls made up of plates measuring 30 mm × 300 mm × 300 mm. In this test, an internal heat source is used to establish a stationary heat flow, which allows a temperature difference to be obtained between the inside and outside of the thermal box. Subsequently, with the help of thermocouples, it is possible to measure the thermal jump produced between the internal and external surface temperature of the gypsum composite, which allows the thermal conductivity coefficient of each material to be calculated;In order to study the effect of water absorption by capillary action on the gypsum composites designed for this research, the recommendations of the RILEM TC 25-PEM standard [31] were applied. The test was carried out using a series of three prismatic specimens of each material measuring 40 mm × 40 mm × 160 mm, immersed vertically in a container with water to a depth of 10 ± 1 mm. The duration of the test is 10 min and allows the height reached by the water in each compound to be determined, as well as the amount of water absorbed for each material designed;Finally, the water vapour permeability of the materials produced was determined according to the recommendations of the UNE-EN ISO 12572 standard [32]. For this purpose, circular specimens with a diameter of 100 ± 1 mm and a thickness of 10 ± 1 mm were placed in a watertight container containing 200 ml of a saturated solution of potassium nitrate. The samples were tested weekly for seven weeks, observing the loss of mass that occurs as a consequence of the evaporation of the salt solution through the plaster compound, which allows the determination of the water vapour permeability coefficient.

For the mechanical characterisation, flexural and compressive strength tests were carried out in accordance with the UNE-EN-13279-2: 2014 standard [29]. For this purpose, a series of three specimens of each RILEM standard dimensions of 40 mm × 40 mm × 160 mm and a hydraulic press model AUTOTEST 200-10SW were used (Figure 5d,e). Additionally, for the discussion of the mechanical results, scanning electron microscopy images were obtained to observe in detail the matrix of the lightened gypsum composites. These images were taken with the aid of a Jeol JSM-820 microscope operating at 20 kV, equipped with Oxford EDX analysis. Prior to imaging, the samples were coated with a thin layer of gold using a Cressington 108 metalliser to ensure good conductivity to the electron beam generated by the equipment.

The last phase of this research consisted of producing prefabricated blocks of original design, measuring 15 cm × 15 cm × 30 cm, especially conceived for the construction of lightened interior partitions for housing. The blocks were designed under circular economy criteria that allowed the recovery of waste containers to build the cells that make up the modular piece, as well as the recovery of waste prefabricated gypsum panels as formwork. These prefabricated units were tested in compression until obtaining the necessary breaking load for cracking using an IBERTEST MIB-60/AM universal press.

## 3. Results

This section includes the tests derived from the physical and mechanical characterisation carried out on the gypsum composites developed for this research, as well as the tests carried out to determine the application possibilities of this novel material for the elaboration of prefabricated interior blocks.

### 3.1. Physical Characterisation Tests

Firstly, Table 5 shows the results obtained for the bulk density and Shore C surface hardness of the gypsum composites used.

Table 5 shows how the surface hardness decreases as the amount of granulated ELT residue incorporated in the matrix of the plaster composites increases, as shown in other studies [33]. On the other hand, the addition of recycled mineral wool fibres does not change this property, presenting higher surface hardness values the specimens made with rock wool fibres. Regarding the bulk density, it can be seen that the composites with the addition of 300 g of ELT aggregates are lighter than those incorporating only 150 g of this waste. It can also be seen that the bulk density of the composites with the addition of recycled fibres is slightly lower than that obtained by the composites incorporating only granulated ELT. These results suggest the potential application of these gypsum composites to produce lightweight prefabricated products that can be used in the interior of dwellings. 

These results suggest the potential application of these gypsum composites to produce lightweight prefabricated products that can be used in the interior of dwellings.

Figure 6 presents the results obtained for the thermal conductivity coefficient of the different gypsum lightweight composites used in this research. This physical property of the building materials is essential to determine their application possibilities, as well as the potential energy savings derived from the use of this novel material in the construction of interior partitions [34]. The incorporation of ELT granulated waste and recycled mineral wool fibres in gypsum composites aims to increase the thermal resistance of traditional building materials, while at the same time recovering raw materials that would otherwise end up in open-air landfills, generating a strong environmental impact [35].

Figure 6 shows how the incorporation of 2.5–4.0 mm granulated rubber waste particles in the mixing process of gypsum composites increases their thermal resistance. This decrease in thermal conductivity is higher in the composites containing 300 g of ELT compared to those with 150 g, in accordance with the results obtained for the bulk density in Table 5. On the other hand, it is observed that the addition of recycled mineral wool fibres has a positive effect on the thermal conductivity of the lightened composites, with a lower thermal conductivity in the composites elaborated with RW compared to those containing GW. The results obtained for the thermal conductivity coefficient are in line with those obtained by other researchers who incorporate synthetic wastes in the gypsum composite matrix, as can be seen in Table 6.

Figure 7 shows the results obtained for the capillary water absorption test, where the height reached by the water in each of the gypsum composites analysed and the mass of water absorbed can be observed.

It is shown in Figure 7 how the incorporation of rubber residues from ELT in the matrix of the gypsum composites leads to a decrease in the capillary rise of water. It can be appreciated that in those dosages with raw material replacement by 300 g of recycled rubber, the water reaches a lower height than in the composites with 150 g replacement. On the other hand, it can also be seen that the addition of recycled thermal insulation fibres has a positive effect by slightly reducing the capillary height reached by the water in these gypsum composites. However, it is observed that the mass of water absorbed by the composites incorporating rubber waste is greater than for the reference plaster, and this absorbed water increases as the content of gypsum replaced by ELT increases. These effects were observed by López-Zaldívar et al., who demonstrated in their research that the open porosity of gypsum composites increases with the incorporation of ELT granular particles, from which it follows that they can absorb more water mass and reach a lesser capillary height [15].

Finally, Figure 8 shows the results of the water vapour permeability test on the gypsum composites produced in this work.

From the analysis of Figure 8, it can be seen that the incorporation of granulated rubber waste in the 2.5–4.0 mm format leads to a decrease in water permeability. This decrease in permeability is greater the more ELT is added to replace the original raw materials in the gypsum composites. A similar effect was observed by Vidales-Barriguete et al. in their research with gypsum composites incorporating plastic waste from electrical cables, which demonstrated the technical feasibility of using these wastes to produce more water-impermeable prefabricated products for wet rooms [40]. In turn, the addition of recycled thermal insulation fibres also led to an improvement by further reducing water vapour transmission through the gypsum composites used in this research.

### 3.2. Mechanical Characterisation Tests

This section presents the results obtained in the flexural and compressive mechanical strength tests for the gypsum composites produced. These tests allow us to determine the possibilities of use of this new material, showing whether they comply with the minimum required by the UNE-EN 13279-2 standard for each of the mentioned strengths (1 MPa in bending and 2 MPa in compression) [29]. The addition of recycled thermal insulation fibres in the matrix is intended to correct the negative effects of replacing the gypsum paste with the granulated rubber residue. However, it is understood that these properties are diminished as a consequence of the decrease in density caused by the incorporation of these residues. The results are presented in Figure 9a,b.

As shown in Figure 9, there is a progressive decrease in the mechanical properties of the composites produced as the amount of rubber aggregates introduced as a partial substitute for the original gypsum material increases. Thus, the dosages with 300 g of ELT show a worse mechanical behaviour than those with 150 g, in accordance with the results obtained for their bulk density and their lower surface hardness. Although in no case are the strengths of the gypsum material without additions (G0.7) improved, all the composites produced exceed the minimum strength values of 1 MPa in bending and 2 MPa in compression set by current standards. On the other hand, Figure 9a shows the beneficial effect derived from the addition of recycled fibres in the matrix of the elaborated materials, significantly increasing the flexural strength of the composites lightened with rubber aggregates. In this sense, there is an increase in flexural strength of more than 20% in all the composites with fibres, compared to the composites lightened without fibres. However, the positive effect of fibres on the increase in compressive strength observed in Figure 9b is not as significant (around 5% with respect to gypsum materials lightened without fibres). Similar effects have been observed in other investigations that developed new gypsum materials reinforced with natural or synthetic fibres [41,42].

To complement the discussion of the mechanical properties studied, Table 7 shows the results obtained by other researchers who worked with gypsum composites lightened with synthetic residues. It can be seen that the flexural and compressive strength values achieved in this work are in line with the results of the aforementioned research.

Finally, Figure 10 shows images obtained by scanning electron microscopy for the sample G07–300–RW, which show the morphology of the matrix of the gypsum composites produced. To obtain the images shown in Figure 10, the secondary electron technique was used, capturing images at different magnifications of the sample obtained after the flexural breakage test at seven days of age.

The images shown in Figure 10 were taken from a sample obtained from the centre of the matrix of a 4 × 4 × 16 cm specimen of the G0.7–300–RW composite. Figure 10a shows the good dispersion of the mineral wool insulation residue fibres over the entire surface of the matrix of the composite under study. In addition, as shown in Figure 10b, these fibres present good adhesion to the gypsum binder, generating dihydrate crystals throughout their interface that hinder fibre sliding and, therefore, brittle breakage. Finally, Figure 10c highlights the good cohesion between the rubber aggregates and the gypsum matrix, which prevents the segregation of these granular particles.

### 3.3. Design and Characterisation of a Precast Block Prototype

This section includes the characterisation of an innovative prototype of a prefabricated block for modular construction of partition walls and interior wall linings. Figure 11 shows the raw materials used for its manufacture, the dimensions and composition of the prefabricated product, as well as its final state and the testing method used. The same techniques and methods described in Section 2.2 were used to prepare the blocks.

Figure 11 illustrates how, for the production of the prefabricated part, the recovery of plastic waste from bottles was used to make the nine-centimetre-diameter cells. These polyethylene terephthalate containers were cut to a height of 15 cm, discarding the narrowing of the bottle, which was reintroduced into the normal recycling process for plastic containers. In turn, waste prefabricated plasterboard panels recovered from construction sites in the Community of Madrid (Spain) were used to form the formwork. These pieces were manually cut to obtain the dimensions 15 cm × 30 cm × 1.2 cm filled. The results obtained for the compressive strength and density of the tested precast elements are shown in Figure 12.

As can be seen in Figure 12, there is a decrease in compressive strength as the density of the precast blocks produced is reduced. Thus, the pieces with the incorporation of 300 g of ELT granular particles in place of the original gypsum material reduce their strength by about 15% more than those composites with the addition of 150 g of rubber aggregates. On the other hand, the incorporation of recycled thermal insulation fibres does not lead to a significant increase in compressive strength, although it does prevent the brittle fracture type of the blocks after testing. Similar results were obtained by Villoria et al. in their research, demonstrating how prefabricated gypsum blocks were a viable alternative to develop interior partitions and wall linings, since, in general terms, the compressive stresses that these prefabricated blocks can reach would not exceed their breaking load as they are non-structural elements [45].

## 4. Conclusions

The development of new, more sustainable, and environmentally friendly building products has become one of the major challenges for the industry to meet the sustainable development goals (SDGs). This work is in line with SDG 12: responsible production and consumption, through the development of a novel gypsum composite that reduces the consumption of raw materials, while encouraging the recovery of ETL and thermal insulation waste for its reincorporation into the manufacturing process of a more sustainable building material. 

Among the most relevant conclusions that can be drawn from the research carried out are the following: The gypsum composites developed show how it is possible to replace up to 300 g of the original raw materials with granulated rubber waste from ELT, which means a saving of more than 17.5% of the binder/water mixture;The incorporation of rubber aggregates in combination with recycled fibre mineral wool insulation reduces the density of hardened gypsum composites, in turn reducing the thermal conductivity of these materials by up to 30%. They are, therefore, presented as an alternative solution to improve the energy efficiency of buildings;On the other hand, the incorporation of these wastes decreases the water vapour permeability and the capillary height reached by water compared to traditional gypsum composites (G0.7). This suggests the technical feasibility of the possible application of these materials as prefabricated materials in wet rooms;Although there is a decrease in the mechanical properties as a consequence of the incorporation of the rubber aggregates and the decrease in the gypsum paste content, in all the cases analysed, the minimum values required by the current regulations for flexural strength (1 MPa) and compressive strength (2 MPa) are exceeded. In addition, the good adhesion of the recycled mineral wool fibres to the gypsum matrix is observed, which results in better flexural behaviour and reduces the risk of brittle fracture of these sustainable construction materials;Finally, the possibility of using these gypsum composites for the production of prefabricated blocks for wall lining and interior partition was studied. These prefabricated products obtain good results in compression for low densities and also make it possible to recover plastic waste from bottles and prefabricated plate waste from construction and demolition works.

In this sense, the lightened gypsum composites designed for this research are presented as a viable alternative for the production of more sustainable prefabricated construction products. This would, in turn, provide a competitive advantage in terms of product differentiation for those companies that wish to offer products with a marked environmental character. In any case, the economic study derived from the real application of these construction materials on site and the savings in transport caused as a result of their low density is proposed as a future line of work.

## Figures and Tables

**Figure 1 materials-16-00243-f001:**
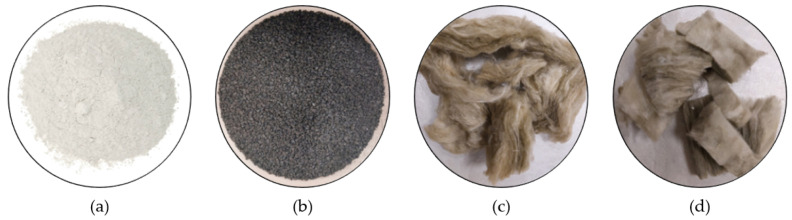
Raw materials used: (**a**) fine gypsum; (**b**) granular end-of-life tyre particles; (**c**) recycled rock wool fibre; and (**d**) recycled glass wool fibre.

**Figure 2 materials-16-00243-f002:**
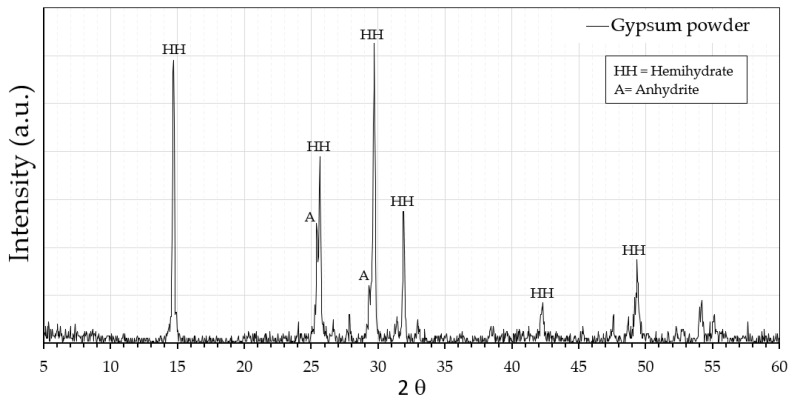
Analysis by X-ray diffraction (XRD) of the gypsum powder.

**Figure 3 materials-16-00243-f003:**
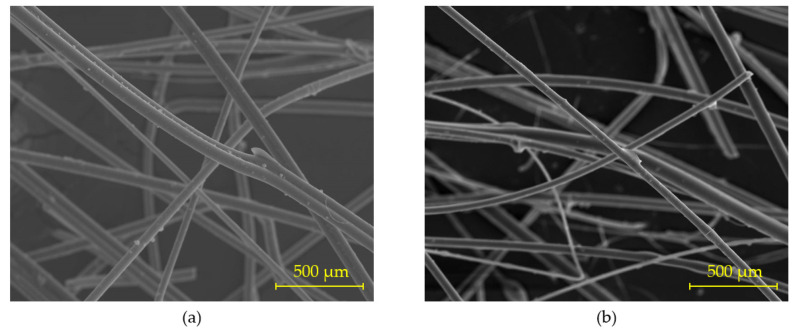
Electron microscopy for recycled fibres obtained by secondary electron imaging: (**a**) rock wool fibre and (**b**) glass wool fibre.

**Figure 4 materials-16-00243-f004:**
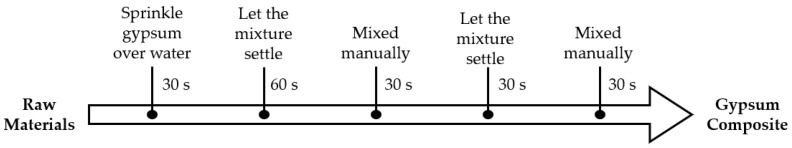
Mixing process of the gypsum compounds elaborated.

**Figure 5 materials-16-00243-f005:**
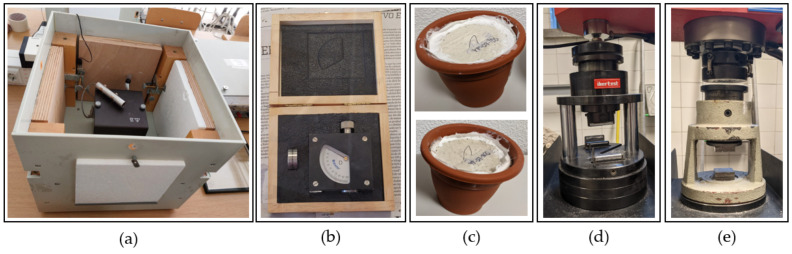
(**a**) Thermal box; (**b**) Shore C hardness tester; (**c**) water vapour permeability test; (**d**,**e**) hydraulic press, flexural and compression test modules.

**Figure 6 materials-16-00243-f006:**
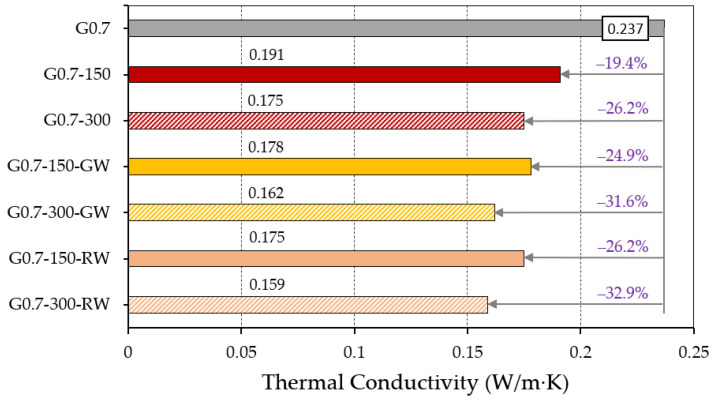
Thermal conductivity coefficient for the different gypsum composites.

**Figure 7 materials-16-00243-f007:**
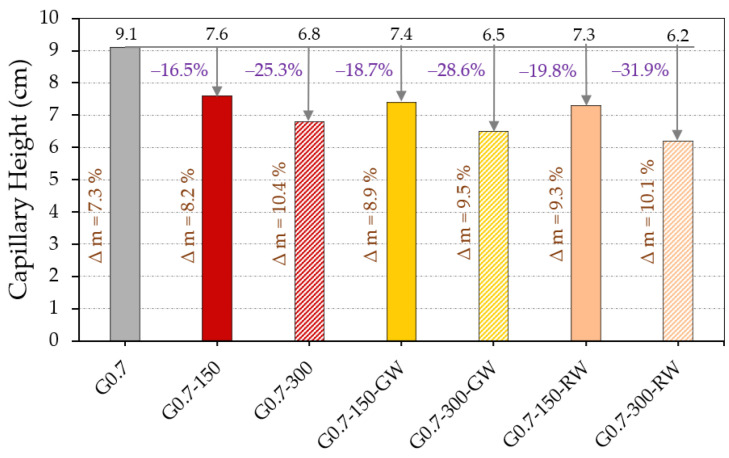
Results of the capillary water absorption test.

**Figure 8 materials-16-00243-f008:**
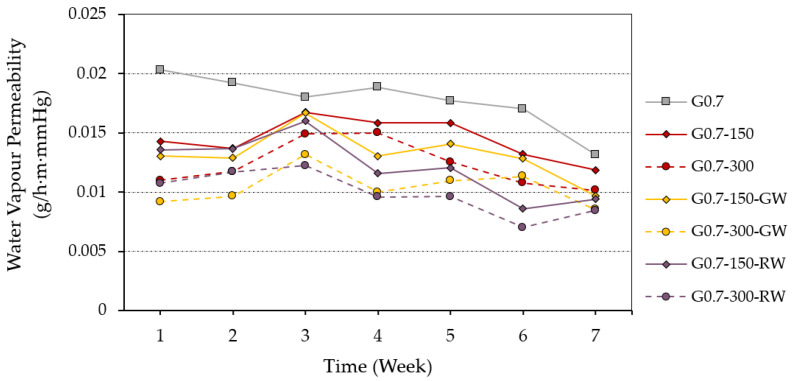
Results of the water vapour permeability test measured over seven weeks.

**Figure 9 materials-16-00243-f009:**
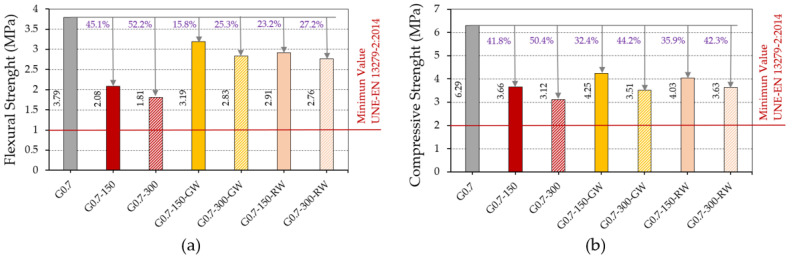
Results of the mechanical characterisation of gypsum composites: (**a**) flexural strength and (**b**) compressive strength.

**Figure 10 materials-16-00243-f010:**
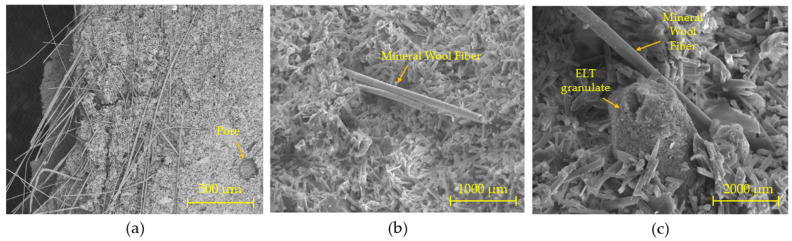
Secondary electron scanning electron microscopy (SEM) images of sample G0.7–300–RW: (**a**) 500 μm; (**b**) 1000 μm; and (**c**) 2000 μm.

**Figure 11 materials-16-00243-f011:**
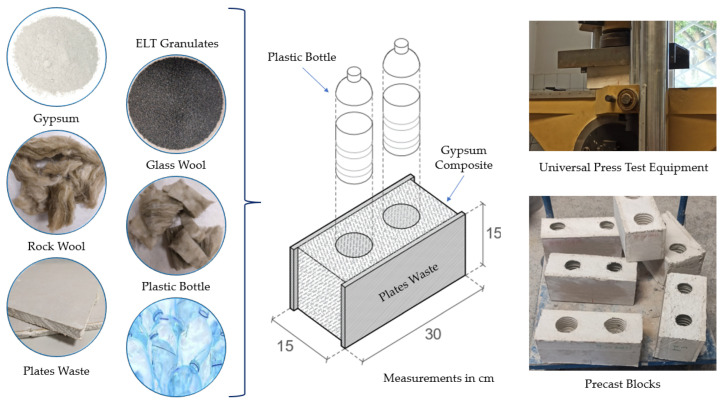
Process of elaboration and testing of the prefabricated blocks elaborated for this research.

**Figure 12 materials-16-00243-f012:**
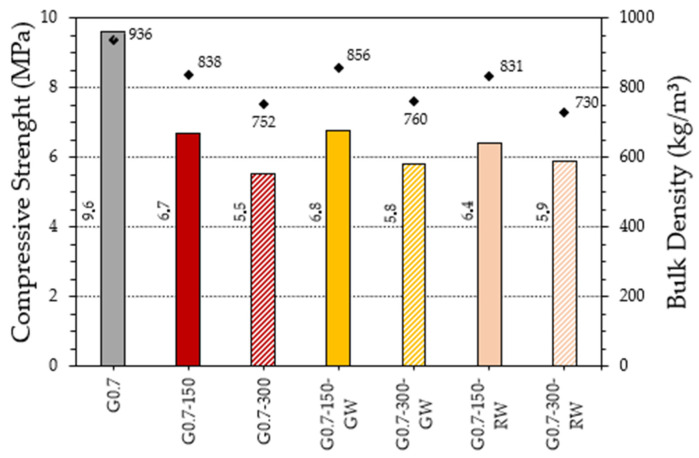
Compressive strength of precast blocks and bulk density.

**Table 1 materials-16-00243-t001:** Physical properties of fine gypsum YF-B1. Source: Saint-Gobain Placo.

Fire Resistance	pH	Particle Size	Water Vapour Diffusion Factor	Workability Time	Purity Rate
Euroclass A1	>6	0–0.4 mm	μ = 6	10–15 min	>80%

**Table 2 materials-16-00243-t002:** Physical properties of recycled crumb rubber from end-of-life tyres.

Particle Size	Grain Morphology	Bulk Density	Humidity (by Weight)	Textile Material Content (by Weight)	Ferromagnetic Materials Content (by Weight)
2.5–4.0 mm	Angular	550 kg/m^3^	<0.75%	<0.50%	<0.10%

**Table 3 materials-16-00243-t003:** Physical properties of the thermal insulation mineral wool waste used. (Source: adapted from URSA IBERICA, S.A.).

Type	Fibre Width (μm)	λ (W/m·K)	Fire Resistance (Euroclass)	Water Vapour Diffusion Factor	Water Absorption (kg/m^2^)
Glass wool	15–20	0.034	A2-s1, d0	<1	<1
Rock wool	8–9	0.035	A1		

**Table 4 materials-16-00243-t004:** Mass ratios used for sample preparation.

Type	Gypsum (g)	Water (g)	Rubber Aggregate (g)	Mineral Wool (g)
G0.7	1000	700	—	—
G0.7–150	912	638	150	—
G0.7–300	824	576	300	—
G0.7–150–GW	910	638	150	2
G0.7–300–GW	822	576	300	2
G0.7–150–RW	910	638	150	2
G0.7–300–RW	822	576	300	2

**Table 5 materials-16-00243-t005:** Shore C surface hardness and bulk density.

Property	G0.7	G0.7–150	G0.7–300	G0.7–150–GW	G0.7–300–GW	G0.7–150–RW	G0.7–300–RW
Surface Hardness (Ud. Shore C)	75	66	61	64	59	68	62
Bulk Density (kg/m^3^)	1108.95	946.33	864.61	935.16	849.22	957.58	852.73

**Table 6 materials-16-00243-t006:** Thermal conductivity obtained in other studies of gypsum composites lightened with synthetic residues.

Residues	ELT Rubber (0.6 mm) [13]	ELT Rubber (2.5 mm) [13]	Cork [36]	Cellular Glass [37]	Plastic Cables [38]	Polycarbonate [39]
λ (W/m·K)	0.197	0.188	0.124	0.310	0.230	0.170

**Table 7 materials-16-00243-t007:** Flexural and compressive strength results obtained in other investigations with gypsum composites as a function of the type of synthetic waste added.

Residues	EPS ^1^ [43]	XPS ^2^ [44]	Cables [38]	Polycarbonate [39]	Mineral Wool Fibres [22]	ELT Rubber (0.6 mm) [13]
Flexural strength (MPa)	1.81	2.32	2.63	3.32	4.11	2.05
Compressive strength (MPa)	3.11	3.15	5.12	7.98	6.97	2.74

^1^ EPS—expanded polystyrene; ^2^ XPS—extruded polystyrene.

## Data Availability

Not applicable.

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
