# Peer review of "Recovery of End-of-Life Tyres and Mineral Wool Waste: A Case Study with Gypsum Composite Materials Applying Circular Economy Criteria"

_materials, 2022, doi:10.3390/ma16010243_

Round 1
Reviewer 1 Report
The topic of the paper is interesting and starts to be more and more important nowadays. Even if the utilization of rubber waste in the gypsum has been studied rather often the combination with mineral wool waste is innovative and brings new information . Nevertheless the paper need to be improved in order to be published in an impacted journal. The main problem of the paper is lack of the real discussion. Mostly the results are only described and no explanation is given. The comparison with results of other authors is not sufficient, because they were obtained for the materials with different compositions and the type of gypsum and amount of water play significant role in their behaviour.
Comments:
Page 2, line 53: "....the rubber granules generated after tyre shredding can be added as a partial replacement of the binder..." : Rubber granules can be used only as a filler in the gypsum matrix, therefore they could not be added as a replacement of binder.
P. 3, l. 107: Equation (2) does not describe the production of building gypsum. Building gypsum (in the form of powder) is calcium sulphate hemihydrate (as can be seen in Fig. 2) and its preparation is described by the Equation (1) only. Equation (2) describes the formation of anhydrite at higher temperatures. Even Equation (1) seems to be redundant, because it is not related to the topic of paper.
P. 4, Table 2: word "weight" starts once in small letter, twice in capital letters.
P. 4, Table 3: How the values of thermal insulation coefficients were obtained? Both types of wool seems to be loose fibers and their thermal conductivity would strongly depend on the compaction. Same for water vapour diffusion factor.
P.5, Figure 4: The material was mixed, not kneaded and also not "removed". Word "remove" means "take away". Was the samples compacted in any way? When the samples were unmolded? What was the age of samples at the time of testing?
P. 7, l. 239-249: It is logical that the bulk density of material with rubber is lower, because bulk density of rubber is about 50% lighter than bulk density of gypsum paste. Therefore there is no need to describe it in such extensive way.
P. 8, Table 6: Comparison of the obtained values with values of other authors does not give sense, because thermal conductivity strongly depends on the amount of filler and water and they were different for each author.
P.8, l. 278 - 290: How can you explain the decrease of capillary height and simultaneous increase of absorbed water in materials with ELT?
P. 9, Figure 8: How can you explain the increase of water permeability between 2 and 3 days in all materials with ELT?
P. 10, Table 7: Same comment as for Table 6.
P. 11, Figure 11: Was the material for blocks mixed manually also? And were the block compacted? Were they dried before testing?
P. 12, Figure 12: Please do not connect the values of bulk density, because they are separate for each material and not related to each other.
Author Response
The topic of the paper is interesting and starts to be more and more important nowadays. Even if the utilization of rubber waste in the gypsum has been studied rather often the combination with mineral wool waste is innovative and brings new information. Nevertheless, the paper needs to be improved in order to be published in an impacted journal. The main problem of the paper is lack of the real discussion. Mostly the results are only described and no explanation is given. The comparison with results of other authors is not sufficient, because they were obtained for the materials with different compositions and the type of gypsum and amount of water play significant role in their behaviour.
The authors would like to acknowledge the reviewer's comments that have significantly improved the quality of the submitted manuscript. All proposed revisions have been taken into consideration.
Comments:
Page 2, line 53: "... the rubber granules generated after tyre shredding can be added as a partial replacement of the binder...”: Rubber granules can be used only as a filler in the gypsum matrix, therefore they could not be added as a replacement of binder.
Indeed, fillers are usually used to save binder in order to obtain a lighter material. As can be seen in Table 4, by adding the rubber granules to the gypsum mix, the amount of binder required to produce the same specimen is reduced compared to the reference specimens without additions. The statement that "the rubber granules can be added as a partial replacement of the binder" refers to the fact that, as the quantity of rubber granules increases, the quantities of water and gypsum decrease proportionally.
P. 3, l. 107: Equation (2) does not describe the production of building gypsum. Building gypsum (in the form of powder) is calcium sulphate hemihydrate (as can be seen in Fig. 2) and its preparation is described by the Equation (1) only. Equation (2) describes the formation of anhydrite at higher temperatures. Even Equation (1) seems to be redundant, because it is not related to the topic of paper.
The authors understand the reviewer's comment and equation (2) has been deleted. Equally, the authors think that Equation (1) is convenient to include, as it gives information about the raw material to people unfamiliar with gypsum in construction.
P.4, Table 2: word "weight" starts once in small letter, twice in capital letters.
The correction has been made in Table 2.
P.4, Table 3: How the values of thermal insulation coefficients were obtained? Both types of wool seem to be loose fibers and their thermal conductivity would strongly depend on the compaction. Same for water vapour diffusion factor.
Table 3 shows the average values for the thermal conductivity corresponding to the commercial thermal insulation boards commonly used in buildings, from which the recycled fibres were obtained. The reference of the commercial brand URSA IBERICA S.A. has been added, and it is highlighted that the final thermal behaviour is studied in gypsum composites reinforced with these recycled fibres.
P.5, Figure 4: The material was mixed, not kneaded and also not "removed". Word "remove" means "take away". Were the samples compacted in any way? When the samples were unmolded? What was the age of samples at the time of testing?
The word 'kneading' has been replaced by 'mixing', thus describing more adequately the process carried out, and Figure 4 has been modified.
As mentioned in the text, "The preparation of the samples used in this research was carried out following the procedure of the UNE-EN 13279-2:2004 standard", this standard specifies that the specimens should be compacted by lifting the mould 10 mm from the top and dropping it 5 times. The specimens were demoulded once the material had set, as indicated in the standard. This information has been added to the text.
The age of samples at the time of testing is described in the text; "After seven days of storage under laboratory conditions the samples were dried in an oven at a constant temperature of 40 ± 2°C for 24 hours prior to testing".
P.7, l. 239-249: It is logical that the bulk density of material with rubber is lower, because bulk density of rubber is about 50% lighter than bulk density of gypsum paste. Therefore, there is no need to describe it in such extensive way.
The paragraph indicated by the reviewer refers to the results of the hardness, the density derived from the addition of rubber aggregates and the effect of the recycled fibres on the weight of the composites. The text only mentions that as the amount of rubber in the composites increases, the density decreases, so the authors consider that this property is not described too extensively.
P.8, Table 6: Comparison of the obtained values with values of other authors does not give sense, because thermal conductivity strongly depends on the amount of filler and water and they were different for each author.
The authors understand the reviewer's comment. However, the intention in adding this table is to show only the mean values obtained in these investigations, in order to guide future researchers interested in the subject as to what results might be expected. The reviewer's comments will be taken into consideration in future work and perhaps a comparative study can be made taking into account the percentage reduction.
P.8, l. 278 - 290: How can you explain the decrease of capillary height and simultaneous increase of absorbed water in materials with ELT?
The phenomenon of capillarity is linked to Jurin's Law, which states that the capillary height reached by water is greater the smaller the diameter of the pore. For this reason, samples with a very large pore size would absorb a greater mass of water, despite the fact that the water would reach a lower height in the sample. This phenomenon has been experimentally proven in other investigations, see the article: https://doi.org/10.1016/j.conbuildmat.2021.125988
P.9, Figure 8: How can you explain the increase of water permeability between 2 and 3 days in all materials with ELT?
Firstly, the time is measured in weeks and not in days. On the other hand, this effect may be caused by water saturation in the gypsum sample after the initial evaporation process. Nevertheless, this is an interesting question that will be taken into consideration for future research, perhaps including DHT-22 type sensors and Arduino equipment to monitor this situation in real time.
P.10, Table 7: Same comment as for Table 6.
This type of tables are indicative and allow the reader to be informed about the behaviour of gypsum composite materials with the addition of residues. This type of work has allowed the development of review articles such as "https://doi.org/10.1016/j.jobe.2021.103338", where it can be seen that the average value is taken as indicated in this work. In any case, the authors will take the reviewer's comments into account for future research.
P.11, Figure 11: Was the material for blocks mixed manually also? And were the block compacted? Were they dried before testing?
The process of preparation and elaboration of the conglomerate material that forms the matrix of the blocks has been the same as that specified in section 2.2 Sample preparation. This information has been added to the text.
P.12, Figure 12: Please do not connect the values of bulk density, because they are separate for each material and not related to each other.
The connections between the densities have been removed in Figure 12.
Reviewer 2 Report
The work is interesting and meaningful, while some issues should be concerned:
1. What is the full name YF-B1 in line 101?
2. The fibre width showed in Figure 3 is not consistent with the table 3.
3. What is the reason for the mass ratio showed in table 4?
4. In table 4, the mass ratio of G0.7-150-GW is the same with G0.7-150-RW, G0.7-300-GW is the same with G0.7-300-RW, so what is the difference between them?
Author Response
The work is interesting and meaningful, while some issues should be concerned:
The authors would like to acknowledge the reviewer's comments which have been taken into consideration to improve the quality of the work presented.
What is the full name YF-B1 in line 101?
According to EN 13279-1, YF-B1 gypsum is referred to as: YF, fine gypsum; B1, building gypsum. This definition is already given in the text, however, the standard has been added to reflect this nomenclature.
The fibre width showed in Figure 3 is not consistent with the table 3.
The information in Table 3 has been extracted from the mineral wool manufacturers as indicated in the text. The microscopies in Figure 3 were carried out on this type of thermal insulation waste and correspond to the materials indicated in Table 3.
What is the reason for the mass ratio showed in table 4?
One of the objectives of the publication of any research article is that it is reproducible by other researchers, so the methods and procedures carried out must be described in detail. For this reason, the dosages used in this research are specified in Table 4, which also shows the difference between the compounds developed and the partial substitution of the binder that has been carried out.
In table 4, the mass ratio of G0.7-150-GW is the same with G0.7-150-RW, G0.7-300-GW is the same with G0.7-300-RW, so what is the difference between them?
As described in the text: "the type of thermal insulation residue recovered, which can be either glass wool (GW) or rock wool (RW)". Therefore, the difference between the dosages is the type of fibre used, which will have different behaviour in the composites depending on their properties.